# Smart low interfacial toughness coatings for on-demand de-icing without melting

Zahra Azimi Dijvejin[1,2], Mandeep Chhajer Jain[3], Ryan Kozak[3], Mohammad H. Zarifi ®[3] ✉ & Kevin Golovin ®[1,2,4] ✉

Ice accretion causes problems in vital industries and has been addressed over the past decades with either passive or active de-icing systems. This work presents a smart, hybrid (passive and active) de-icing system through the combination of a low interfacial toughness coating, printed circuit board heaters, and an ice-detecting microwave sensor. The coating's interfacial toughness with ice is found to be temperature dependent and can be modulated using the embedded heaters. Accordingly, de-icing is realized without melting the interface. The synergistic combination of the low interfacial toughness coating and periodic heaters results in a greater de-icing power density than a full-coverage heater system. The hybrid de-icing system also shows durability towards repeated icing/de-icing, mechanical abrasion, outdoor exposure, and chemical contamination. A non-contact planar microwave resonator sensor is additionally designed and implemented to precisely detect the presence or absence of water or ice on the surface while operating beneath the coating, further enhancing the system's energy efficiency. Scalability of the smart coating is demonstrated using large (up to 1 m) iced interfaces. Overall, the smart hybrid system designed here offers a paradigm shift in de-icing that can efficiently render a surface ice-free without the need for energetically expensive interface melting.

Undesired ice accumulation is problematic in industries such as renewable energy (wind turbines[1,2], hydroelectric dams[3]), aviation[4], and power transmission[5]. Ice mitigation strategies can be divided into either active or passive methods. Active de-icing involves an external energy input used to remove the ice, typically through thermal, chemical, or mechanical methods. In contrast, passive de-icing either reduces the accretion rate of ice, lowers the adhesion strength between ice and the surface, or both. Neither route towards an ice-free surface is seen as a cure-all today, as active de-icing methods utilize substantial energy but passive de-icing coatings cannot keep a surface ice-free indefinitely. A hybrid system that synergistically combines passive and active de-icing

technologies may be an attractive solution to the ice accretion paradigm.

Electrical devices have been widely employed for active de-icing on a variety of surfaces[6–8] and utilize joule heating to raise the temperature of the accreted ice above 0 °C, facilitating its removal through a phase change to liquid water[9–12]. Proper thermal/electrical conductivity is required to maximize the de-icing efficiency while minimizing energy consumption[9,13,14]. Graphene-based heaters[6,15], hot air pumping[16], conductive polymer-based heaters[17–19] and, most commonly, metallic heating systems[20–23] have all been used to provide sufficient heat to melt the interfacial ice. For example, Bustillos et al. fabricated a highly thermally/electrically conductive and flexible graphene foam heater

[1]Okanagan Polymer Engineering Research & Applications Laboratory, School of Engineering, University of British Columbia, Kelowna, BC V1V 1V7, Canada. [2]Department of Mechanical & Industrial Engineering, University of Toronto, Toronto, ON M5S 3G8, Canada. [3]Okanagan Microelectronics and Gigahertz Applications (OMEGA) Lab, School of Engineering, University of British Columbia, Kelowna, BC V1V 1V7, Canada. [4]Department of Materials Science & Engineering, University of Toronto, Toronto, ON M5S 3G8, Canada. ✉e-mail: mohammad.zarifi@ubc.ca; Kevin.golovin@utoronto.ca

that could raise the interface temperature from −20 °C and start to melt a frozen droplet within 33 sec[19]. Rahimi et al. used plasma spray to deposit NiCrAlY on a glass/epoxy composite, and showed that both fine and rough morphologies could produce sufficient heat for de-icing purposes[23]. Another active de-icing method used by the aviation industry involves flowing hot bleed engine air through the wings of aircraft. Pellissier et al. characterized such hot air pumping for de-icing and their simulation results show that the heat transfer process is highly complex[24]. However, all previous active de-icing techniques, though effective, have required the entire interface to be raised above 0 °C, and accordingly these methods consume considerable energy to de-ice large surfaces such as wind turbine blades, aircraft wings, or boat hulls.

As an alternative, passive de-icing methods utilize coatings with specific surface properties to lessen the ice accretion rate or reduce ice's adhesion to the surface so it can be removed by its own weight, wind, or other aerodynamic/environmental forces. In their recent review, Dhyani et al. detail the many surface design strategies for passive de-icing[25]. In terms of ice accretion delay, superhydrophobic surfaces (SHS) are known for their excellent water repellency with high water contact angle and low contact angle hysteresis[26]. SHS have demonstrated good laboratory scale de-icing in terms of icing delay, removal of supercooled water droplets, and droplet freezing delay due to their low thermal conductivity and minimal surface/droplet contact area[27–29]. However, the icing delay of SHS is typically measured on the scale of minutes, still necessitating a method of ice removal once it has accreted.

Passive de-icing coatings can also reduce the adhesive bond between ice and the coated substrate without necessarily reducing the ice accretion rate. Polydimethylsiloxane (PDMS) and polytetrafluoroethylene (PTFE) are two materials known for their low surface energy that frequently have been used for passive de-icing[30]. Due their weak bond with ice, such materials have shown exceptionally low ice adhesion strengths in various coating configurations including thin films[31], self-assembled monolayers[32], and lubricant infused surfaces[33,34]. Zhao et al. fabricated silicon-oil infused icephobic coatings that demonstrated a low shear ice adhesion strength for cylindrical ice at −10 °C[33]. Similarly, Liu et al. presented fluorinated PDMS films for significant ice adhesion reduction and delayed icing[35]. High surface energy amphiphilic materials can also reduce a surface's ice adhesion strength substantially by creating a nanometer-scale liquid water surface layer, as the bond between liquid water and solid ice is much weaker than a solid-solid bond[34]. For example, the absorption of water vapor has been demonstrated for poly(ethylene glycol) blended with PDMS, resulting in a thin water layer that enhances passive de-icing[36].

Hybrid methods that combine active de-icing and SHS coatings have recently been explored. Cheng et al. fabricated a SHS coating using magnetic particles for hybrid de-icing and showed that increasing the temperature above 0 °C allowed the coating to exhibit excellent ice/water removal[37]. Ma et al. introduced a titanium nitride/polytetrafluoroethylene composite SHS coating as a photothermal de-icing approach[38]. The designed photothermal superhydrophobic surface not only delayed ice formation but also converted absorbed light to heat energy and melted the surface ice. Additionally, Gao et al. demonstrated the use of a hybrid SHS coating and electrical heating for wind turbine de-icing[39]. They showed significant energy savings (90%) when de-icing the whole turbine blade by only coating the leading edge with their SHS coating and electrical heaters. Many other works have demonstrated hybrid ice mitigation combining a SHS and active heating[40–43]. Hybrid de-icing methods can also employ lubricant infusion. Jamil et al. used silicone lubrication on a candle soot coating as a natural light absorber[44]. In their work, conductive iron oxide nano particles served as a heat dissipator, eventually melting the iced interface. However, an intractable issue remains with previous hybrid de-icing approaches.

Because hydrophobic coatings repel only liquid water, melting the ice is required for this strategy to be effective. And so, while the energy consumption in these studies was reduced in comparison to a purely active de-icing method, the required energy was still substantial and would scale with the size of the iced interface. Considering that the latent heat of melting ice (334 J/g) is about 160x greater than ice's specific heat capacity (2.09 J/g °C), a hybrid de-icing system that could avoid melting would provide substantial energy efficiency benefits.

Materials exhibiting low interfacial toughness (LIT) with ice represent a paradigm shift in how the adhesion between ice and a surface may be reduced, especially large (>cm) iced interfaces[45,46]. LIT materials minimize the strain energy necessary to propagate an interfacial crack between the ice and surface, enabling size-independent de-icing, i.e. requiring a constant applied force for ice removal irrespective of the size of the iced interface. To-date, various LIT materials have been reported, including polymers such as polypropylene, PTFE, and ultra-high molecular polyethylene (UHMW-PE)[46], as well as aluminum-based quasicrystalline coatings[45]. Zeng et al. introduced a LIT coating comprised of porous PDMS that exhibited lower interfacial toughness and hydrophobicity with increasing porosity[47]. Dhyani et al. fabricated transparent LIT PDMS and polyvinylchloride (PVC) coatings for photovoltaic applications, simultaneously demonstrating both a low interfacial toughness and ice adhesion strength[48]. Yu et al. fabricated robust LIT coatings based on PTFE particle assemblies, where the interfacial toughness was maintained after repeated icing and de-icing cycles[49]. And yet, to-date LIT materials have only been used as passive de-icing coatings.

In this work we develop hybrid LIT de-icing coatings based on UHMW-PE. Whereas all previous hybrid de-icing technologies have required energy-intensive ice melting, incorporating LIT materials enables mechanical de-icing that circumvents the melting step. The mechanical properties of both the LIT coating and ice determine the toughness and strength of their adhesive interface. Accordingly, the influence of the elastic modulus on ice adhesion strength and interfacial toughness is first measured at varying temperatures (−5 °C to −60 °C) for both the LIT material and ice. Next, we study the effect of thermal loading on the interfacial toughness using miniature printed circuit board resistive heaters. A comprehensive study using several lengths of ice is carried out to optimize the voltage required to increase the surface temperature to −5 °C, where the lowest interfacial toughness with ice was observed. The effect of the supplied heat on the coating's interfacial toughness with ice is investigated by applying the optimal voltage. The coatings are further rendered 'smart' through the inclusion of an embedded microwave resonator sensor, enabling on-demand de-icing where the active system can be shut off immediately once the sensor detects that the surface has been de-iced. The microwave sensor consists of a split ring resonator and transmission lines and operates by exploiting the large difference in dielectric properties between water and ice, as previously demonstrated[50–52]. At the applied optimal voltage, the sensor's response to the presence and absence of ice is also recorded.

## Results and discussion
### Thermomechanical interfacial properties
Before altering the temperature using an active de-icing system, the mechanical properties of the ice and LIT coating were first investigated to understand how they are affected by temperature, either directly or indirectly (for example, due to a change in elastic modulus). Both the interfacial toughness and ice adhesion strength depend on the mechanical properties of the coating and ice. The effect of temperature on the dynamic elastic modulus of polycrystalline ice has been previously measured to follow[53],

$$E = [.104(1 + 1.07 \times 10^{-3}T + 1.87 \times 10^{-6}T^2)]^{-1} \pm 1\% \quad (1)$$

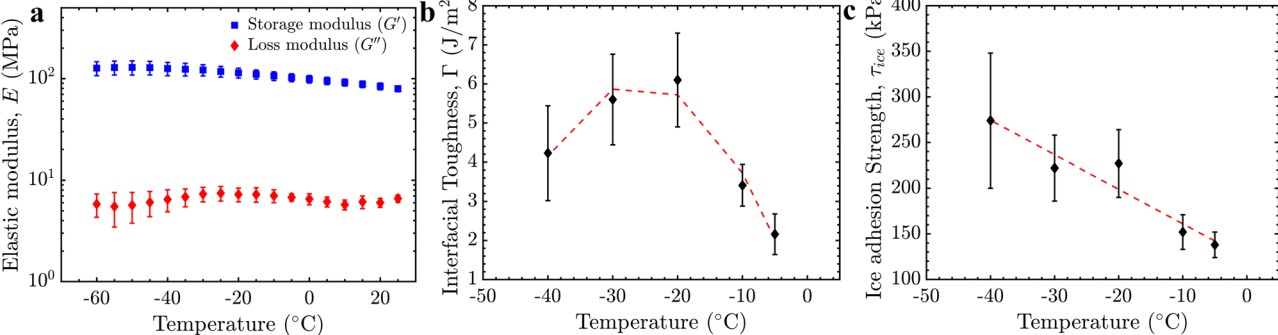

Fig. 1 | **The effect of temperature on the mechanical properties of the coating and its interface with ice. a** Storage (G') and loss (G") moduli of UHMW-PE between 25 °C and −60 °C. **b** The toughness of the ice-UHMW-PE interface (Γ) between −40 °C and −5 °C. **c** The ice adhesion strength ($\tau_{ice}$) of UHMW-PE between −40 °C and −5 °C. Values for strength and toughness were taken from the linear and plateau regions of ice length versus removal force plots (Methods) as described previously[46]. Complete datasets are found in Supplementary Fig. S1. Errorbars represent one standard deviation (SD) and N > 4. Source data are provided as a Source Data file[67].

Here $E$ is the elastic modulus of ice in GPa and $T$ is the temperature in °C. Based on this equation, the modulus of ice decreases about 5% as the temperature increases from −40 °C to −5 °C. The elastic modulus of the coating was investigated using Dynamic Mechanical Analysis (Methods). The loss modulus was statistically constant between 25 °C and −60 °C (Fig. 1a), whereas the storage modulus increased 22% when the temperature decreased from −5 °C to −40 °C. Accordingly, neither the ice nor the coating's mechanical properties varied substantially over the thermal range investigated here, and did not significantly affect the observed interfacial toughness and ice adhesion strength values discussed below.

The interfacial toughness and ice adhesion strength were clearly affected by temperature (Fig. 1b, c). As can be seen in Supplementary Fig. S1, the asymptotic force required to remove large-scale ice increased from $F_c$ = 145 N/cm at −5 °C, to $F_c$ = 237 N/cm at −30 °C. This corresponds to an increasing interfacial toughness with ice from Γ = 2.2 ± 0.5 J/m² to Γ = 5.6 ± 1.2 J/m² over the same thermal range. However, at −40 °C cohesive detachment was observed where a portion of the ice remained on the surface after partial interfacial fracture. While the critical ice removal force decreased to 207 N/cm (statistically different from −30 °C and −20 °C, based on a Student's *t*-test), with cohesive fracture this is no longer a true interfacial property. As discussed above, ice is at most 5% stiffer at the lower temperatures investigated here. This should have decreased the interfacial toughness, which was instead 2.2 times larger at −30 °C than at −5 °C. The ice adhesion strength was similarly 1.8 times greater (Fig. 1b, c). Accordingly, the increases in interfacial strength and toughness cannot be explained as an indirect effect due to the change in mechanical properties of the interface. Instead, it would appear that both properties increase with decreasing temperature, at least for the UHMW-PE/ ice interface investigated here. This increase has been previously reported, but only for the ice adhesion strength, by several groups[54–59].

### Joule heating voltage optimization
With the thermal properties of the ice/LIT coating well-characterized, active de-icing was then studied using the printed circuit board (PCB) heaters (Supplementary Fig. S2). Initially, the heater was operated at 10 V, and a 20 mm long portion of ice above the LIT-coated heater (equal to the size of the heaters, see Supplementary Fig. S3a) was melted. When trying to dislodge this partially melted ice, cohesive fracture and ice shattering occurred (Supplementary Fig. S2b). Accordingly, completely melting the ice adhered directly above the heater was actually detrimental to the performance of the LIT coating by causing cohesive fracture.

Given the results above, the operating voltage was then optimized so as to raise the interfacial temperature above the heater, but keep it

<0 °C. The Peltier stage held the overall system temperature at $T$ = −25 °C while the PCB heater was used to increase the temperature of the interface between a 150 mm long piece of ice and the UHMW-PE (Fig. 2). To optimize the voltage for efficient thermal control, 0.5 V was initially supplied to the heater with a gradual increment of 0.5 V until the desired surface temperature was achieved after 30 s with the heaters on. Three temperature probes were installed to monitor the changes in the ice and interface temperatures (Supplementary Fig. S3). The first probe measured the temperature at the surface of the heater/ LIT coating ($T_H$). Two other temperature probes both measured the temperature inside the ice adhered to the LIT coating ($T_{ice}$), either directly above the heater or 5 cm away along the length direction (Fig. 2a).

To confirm that the heating was localized, first $T_{ice}$ directly above the heater and 5 cm away were compared (Fig. 2a). $T_{ice}$ 50 mm away from the heater remained relatively constant for all input voltages tested, increasing at most 4 °C using 4.4 V. $T_{ice}$ directly above the heater increased with increasing voltages above 1.5 V, and at 4.4 V reached the desired temperature of −5 °C within 30 s of heating. For a piece of ice measuring 150 × 10 × 5 mm, 4.4 V over 30 s increased both $T_{ice}$ and $T_H$ from −25 °C to −5 °C (Fig. 2a, b). The rate of change in $T_{ice}$ and $T_H$ were statistically equivalent at 5.0 ± 0.2 °C/V and 4.9 ± 0.2 °C/V, respectively. Decreasing the length of the ice from 150 mm to 20 or 60 mm lengths of ice did not affect these results (Fig. 2c, d), and accordingly 4.4 V was used as the de-icing voltage for the remainder of this work.

### Using heaters to control interfacial toughness
The coating's interfacial toughness with ice, Γ, is temperature dependent (Fig. 1b), and the PCB heaters can be used to control the temperature of the interface (Fig. 2). Accordingly, we investigated if the heaters could lower the interfacial toughness, using the 4.4 V optimized above. The force per width necessary for ice removal, $F_{ice}$, with the heater operating at 4.4 V was measured for lengths of ice greater than 50 mm, well within the toughness-controlled regime of fracture (see Supplementary Fig. S1). Two experiments were performed, one at $T$ = −20 °C with the heater set to $T_H$ = −5 °C, and the other at $T$ = −30 °C and $T_H$ = −10 °C. For both experiments, the $F_{ice}$ values corresponded much more closely with the values recorded when the entire system was held at $T_H$ rather than $T$ (Fig. 3a, b). For example, for an iced interface length of 150 mm, $F_{ice}$ = 290 ± 50 N/cm at −20 °C and $F_{ice}$ = 172 ± 15 N/cm at −5 °C (Supplementary Fig. S1). For $T$ = −20 °C and the heater set to $T_H$ = −5 °C, the de-icing force was $F_{ice}$ = 157 ± 30 N/cm, statistically equivalent to the $T$ = −5 °C value. Similar results were observed for other lengths of ice, as well as when using $T$ = −30 °C and $T_H$ = −10 °C (Fig. 3b). Accordingly, the PCB heaters could modulate the

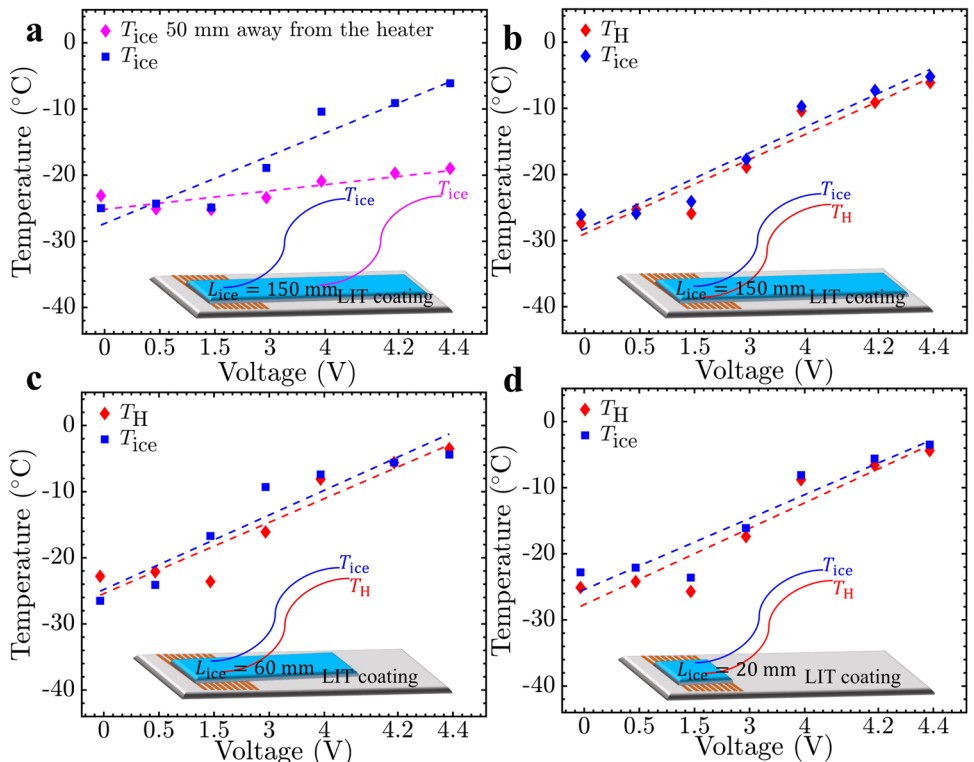

**Fig. 2 | Operational power supply voltage optimization at $T = -25\,°C$. a** Ice temperature, $T_{ice}$, as a function of PCB heater voltage, directly above the heater and 50 mm away along the length direction. The temperature of the interface above the heater ($T_H$) and the temperature of the ice above the low interfacial toughness (LIT) material and heater ($T_{ice}$) were monitored for various lengths of ice: **b** 150 mm, **c** 60 mm, and **d** 20 mm. The target temperature of $T_H = T_{ice} = -5\,°C$ was reached after 30 s at 4.4 V for all three lengths of ice. Source data are provided as a Source Data file[67].

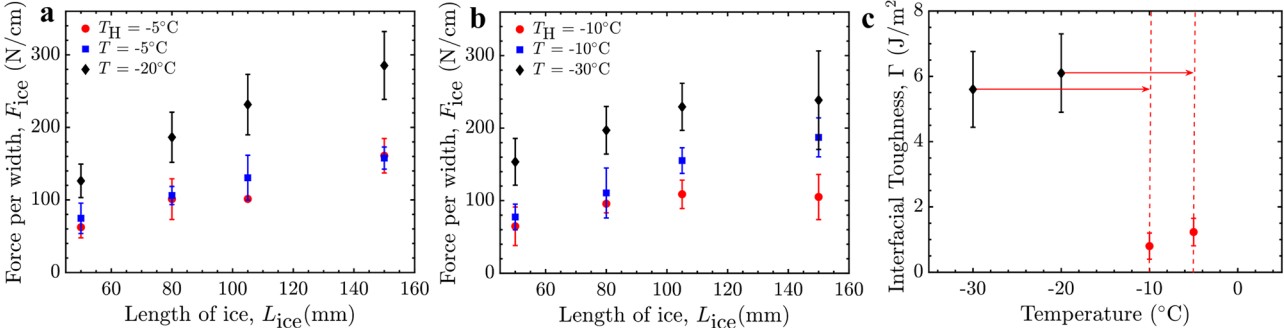

**Fig. 3 | Using the PCB heater to reduce the de-icing force, $F_{ice}$, by locally increasing the temperature from $T$ to $T_H$, for different lengths of ice. a** $T = -20\,°C$ increased to $T_H = -5\,°C$. **b** $T = -30\,°C$ increased to $T_H = -10\,°C$. **c** The effective reduction in interfacial toughness by locally increasing the temperature from $T$ to $T_H$ using the PCB heaters. $T$ is the overall system temperature and $T_H$ is the temperature at the surface of the heater/LIT coating. Errorbars denote 1 SD and here N ≥ 5. Source data are provided as a Source Data file[67].

de-icing force and decreased the interfacial toughness from $\Gamma = 4.8\,J/m^2$ to $1.3\,J/m^2$, and from $\Gamma = 5.7\,J/m^2$ to $0.9\,J/m^2$ (Fig. 3c), for the experiments shown in Fig. 3a, b, respectively. This active de-icing system therefore has the capability to turn a high-toughness interface into a LIT interface on-demand, facilitating large-scale ice removal without melting.

As toughness-mediated fracture is an energy release process, it is likely that increasing the temperature locally compensated for the energy release required for fracture propagation at the interface. As this thermal difference increases, a lower external load is required to propagate fracture. This is in good agreement with the results in Fig. 3, where the interfacial toughness was reduced more when a larger thermal shift was initiated using the PCB heaters. Recall that,

for toughness-mediated interfacial fracture, $F_{ice} = \sqrt{\Gamma E H_{ice}}$, where $H_{ice}$ is the thickness of the ice[60]. Using the heater to locally increase the interfacial temperature from $T = -30\,°C$ to $T_H = -10\,°C$, the de-icing force was measured for ice thicknesses between $H_{ice} = 5-20\,mm$ (Fig. 4a). Here a representative ice length of $L_{ice} = 105\,mm$ was used, well within the toughness-controlled fracture regime (Supplementary Fig. S1). The square root dependence between ice thickness and the measured de-icing force was maintained (Fig. 4b), indicating that the mechanics of fracture were not substantially altered when using the PCB heaters even though the temperature at the interface was nonuniform. Thermal imaging provided additional evidence of this nonuniformity (Fig. 4a), further corroborating that the heating was localized (Fig. 2a) and that

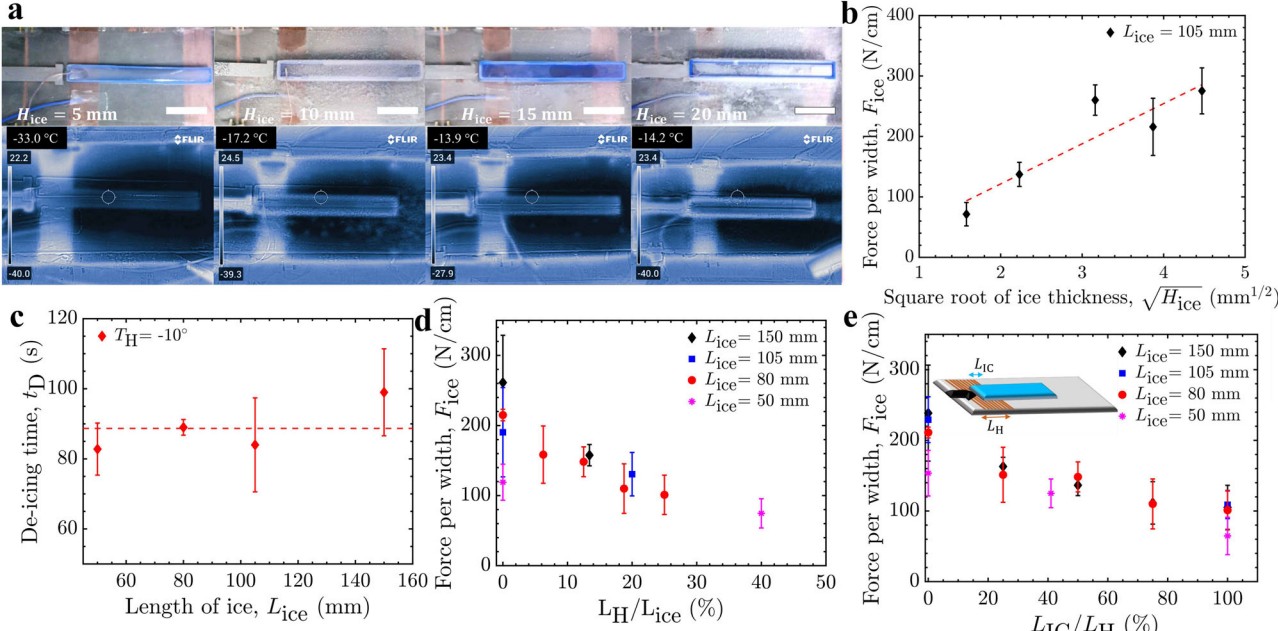

**Fig. 4 | Active LIT de-icing geometry optimization. a** Infrared images demonstrating that locally $T_H$ (the temperature at the surface of the heater/LIT coating) was achieved regardless of the ice thickness. All the scale bars are 25 mm. **b** The square root dependence of the de-icing force with the thickness of the ice. **c** Measured de-icing time for various lengths of ice at system temperature $T = -30\,°C$. The de-icing force of the LIT material as a function of the percentage of: **d** the heater length over the length of ice ($L_H/L_{ice}$), and **e** the length of ice above the heater over the total heater length ($L_{IC}/L_H$). Errorbars denote 1 SD and here $N \geq 5$. Source data are provided as a Source Data file[67].

the temperature of the ice nonadjacent to the heater maintained the colder, ambient temperature.

Given that the entire interface does not need to be heated to improve the LIT properties, an important design parameter is then the required number of heaters and their size and spacing along the interface. We investigated what length of heater, $L_H$, was necessary to decrease the interfacial toughness for various ice lengths, $L_{ice}$. Again 4.4 V was applied over 30 s, and the force required to dislodge the ice was recorded at $T = -30\,°C$. As the portion of ice above the heater ($L_H/L_{ice}$) decreased, the required de-icing force increased (Fig. 4d). However, this was most significant for $L_H/L_{ice} \leq 10\%$, and beyond this the decrease in $F_{ice}$ was minimal. Accordingly, heaters occupying only 10% of the total interfacial area are sufficient for effective de-icing. As expected, when the length of the ice above the heater was equal to the size of the heater ($L_{IC}/L_H = 100\%$), a maximum decrease in the de-icing force was observed (Fig. 4e). However, to minimize power consumption, smaller heaters would be desirable. A statistically insignificant increase in the measured force of ice detachment was observed when the percentage of the heater covered was reduced to 25%. Accordingly, small heaters placed sparsely will still result in good LIT properties while minimizing power consumption. For our laboratory conditions, installing 15 mm long heaters every 135 mm was optimal in order to maximize de-icing while minimizing the power consumption.

The heat flux required for modulating the interfacial temperature would be $Q = U^2 R^{-1} t_D$, where $Q$ is the heater's electrical energy consumption, $U$ is the supply voltage, $R$ is the electrical resistance of the heater, and $t_D$ is the de-icing time[8,61,62]. For our de-icing experiments the operating voltage ($U = 4.4\,V$) and heater resistance ($R = 5.9\,\Omega$) were constant, resulting in $Q = 3.28\,t_D$. The $t_D$ for each length of ice was measured and statistically analyzed (t-test), which revealed that for interfacial lengths between 50 mm and 150 mm, the de-icing time remained constant (Fig. 4c). These results further support that the generated heat was localized and only increased the temperature of the length of ice directly above the heater. According to the recorded average de-icing time ($88 \pm 9\,s$), the consumed electrical energy was

measured as 289 J. Note that the additional mechanical power needed to fracture the interface ($\Gamma A \approx 1\,mJ$) is low and would arise naturally from environmental forces such as wind, drag, or centripetal acceleration (in the case of wind turbines).

The areal power density is a commonly used metric to compare the efficacy of de-icing systems. Previously reported aircraft de-icing systems have required 10–25 kW/m² to achieve ice-free surfaces, with heaters covering the entire iced area[63–65]. For our designed de-icing system the heaters cover only 10% of the surface, which decreases the power consumption by an order of magnitude. Moreover, the heater's resistivity increases with the length of the printed copper on the substrate as, $R = l/\sigma a$ ($\sigma = 5.8 \times 10^8$ S/cm, $l = 2.5$ m, and $a = 1.08 \times 10^{-8}$ m²). Using the 4.4 V de-icing supply voltage, the power density of our heater is $W = U^2 R^{-1} A^{-1} = 2$ kW/m² ($A$ is the area covered by the heater). Accordingly, not only do the heaters cover just 10% of the total area, their resistivity is 10× lower, overall resulting in heaters with a power density 100× higher than the same heaters fully covering the surface. However, this is still an underestimate in the total efficiency gains as our heaters raise 10% of the interface to a subzero temperature, which would not de-ice even for complete heater coverage as it would need to bring the surface to at least 0 °C, and typically much higher[63–65].

## Durability and scalability

In order for our designed hybrid de-icing system to find real-life usage, its performance must be consistent, durable, and scalable. In terms of consistency, the system was exposed to 43 repeated cycles of icing/de-icing. Initially, the ice detachment force for different ice lengths within the toughness regime ($L > L_c$) was measured while the heater locally raised the interfacial temperature from −20 °C to −5 °C (Fig. 5a). The critical detachment force for this first set was 131 ± 21 N, corresponding to an interfacial toughness with ice of $\Gamma = 1.5 \pm 0.4$ J/m². Additional icing/de-icing cycles were then conducted using 150 mm lengths of ice, followed by a repeat of the initial characterization. After these 43 icing/de-icing cycles, the average de-icing force was statistically equivalent (p-value: 0.22) to its initial value. The surface roughness was also

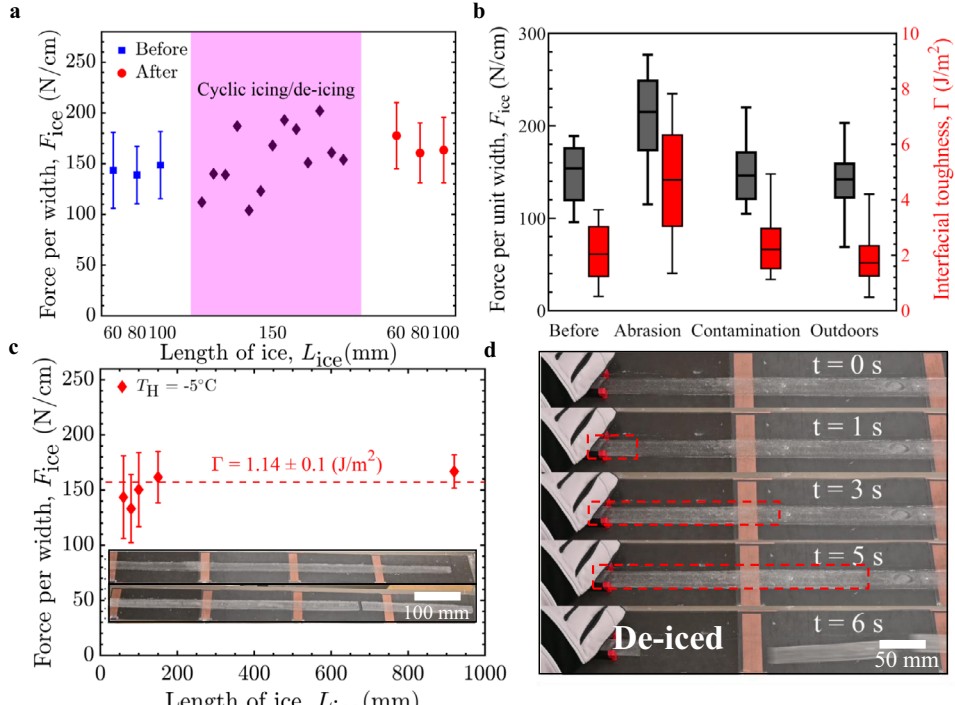

**Fig. 5 | Durability and scalability of hybrid de-icing system. a** Cyclic icing/de-icing tests for ice lengths, $L$, of 60–150 mm ($L > L_c$). $L_c$ is the critical length of ice. De-icing force per width ($F_{ice}$) values before and after the icing/de-icing cycles are statistically equivalent ($p$-value: 0.22). **b** The de-icing force or interfacial toughness ($\Gamma$) required to remove various lengths of ice after mechanical abrasion, chemical contamination, and outdoor exposure for 3 weeks. Minimum and maximum values are shown as the lowest and highest whiskers, respectively. The box presents the first quartile, mean, and the third quartile, from lower to higher amounts. **c** De-icing force for the multi-heater hybrid de-icing system, up to a length of 920 mm. The inset shows the accreted and de-iced surfaces. **d** Movie stills depicting the interfacial crack propagation and adhesive fracture underneath ice with a length of 500 mm and a width of 2 cm. All tests in **a**–**d** were conducted with 2.54 cm wide heaters locally raising the temperature from −20 °C to −5 °C. Errorbars denote 1 SD and here $N \geq 5$. Source data are provided as a Source Data file[67].

unaffected (Fig. S4), indicating that the process of icing and de-icing did not damage the surface.

The de-icing force for various ice lengths in the toughness-controlled regime of fracture was also recorded after mechanical abrasion, chemical contamination, and outdoor exposure for 3 weeks (Fig. 5b, see Fig. S5 for full datasets). The UHMW-PE coating maintained its low ice detachment force for both outdoor exposure and surface contamination ($p$-value > 0.22), demonstrating the environmental durability of the hybrid de-icing system. Only the harsh abrasion increased the de-icing force statistically significantly ($p$-value: 0.002; Fig. 5b). This was due to the increase in roughness of the LIT material, from $S_q = 1.55\,\mu m$ to $3.39\,\mu m$, and this was statistically significant (Fig. S4). As interfacial toughness represents a strain energy per unit surface area, the commiserate increase in toughness with roughness was expected. Note, though, that the increase in interfacial toughness observed while using the heaters to modulate the local interfacial temperature of the abraded UHMW-PE ($3.4 \pm 0.9\,J/m^2$) was still substantially less than that of the unabraded UHMW-PE film without heaters ($6.1 \pm 1.2\,J/m^2$, see Fig. 3c). Accordingly, the hybrid de-icing system can compensate for any mechanical damage by using the heaters to achieve the required toughness value for a given set of environmental conditions.

One advantage of de-icing using LIT materials is their scalability, as the de-icing force is constant for large lengths of ice[45,46]. To determine if our hybrid de-icing strategy was also scalable we fabricated a larger-scale system measuring a full meter in length, and utilizing multiple heaters spaced periodically such that only 10% of the surface area was heated (in-line with the results from Fig. 4d, e). The entire setup was then placed inside a walk-in freezer held at −20 °C and iced using a 2-cm-wide piece of ice (Fig. S6). The de-icing force was measured for a 920 mm length of ice when the heaters modulated the local

interfacial temperature to $T_H = -5\,°C$. Figure 5c shows the de-icing force as a function of ice length for our hybrid de-icing system with heaters underneath only 10% of the total iced area. The de-icing force necessary to dislodge the large-scale ice was statistically equivalent to the values observed for the smaller-scale testing ($p$-value: 0.08), confirming the scalability of the developed system. Scattering of the light between the LIT material and ice during interfacial separation also allowed us to monitor the crack propagation in real time (Fig. 5d). After ~6 s the stored strain energy within the interface was released, and the surface was cleanly de-iced with no adhered residue remaining.

## Smart active de-icing using a microwave resonator sensor

Another consideration in the design of an efficient hybrid LIT de-icing system is to determine the necessary duration of heater usage. A 'smart' system could be envisaged through the addition of an ice sensor which could provide environmental information indicating when to switch the heaters on and off. The smart LIT de-icing system was realized using an embedded microwave sensor (Methods), based on the previously reported work by Kozak et al.[46]. The sensor operates by detecting the change in its resonant amplitude and/or frequency in the presence of ice or water, and was first optimized using finite element method simulations (Supplementary Fig. S7). Once optimized computationally and fabricated experimentally, the effect of the LIT coating on the sensor's response was investigated. After depositing the UHMW-PE on the sensor, the resonant frequency shifted downward by 97 MHz, and the resonant amplitude changed by 1.18 dB, resulting in a resonant peak at 1.908 GHz and −14.73 dB, as shown in Fig. 6a. This shift in the response of the sensor was expected as the LIT material has a dielectric constant of ~2 at 2 GHz (Supplementary Fig. S8) and a small loss factor, which caused the resonant frequency change while minimally altering the resonant amplitude.

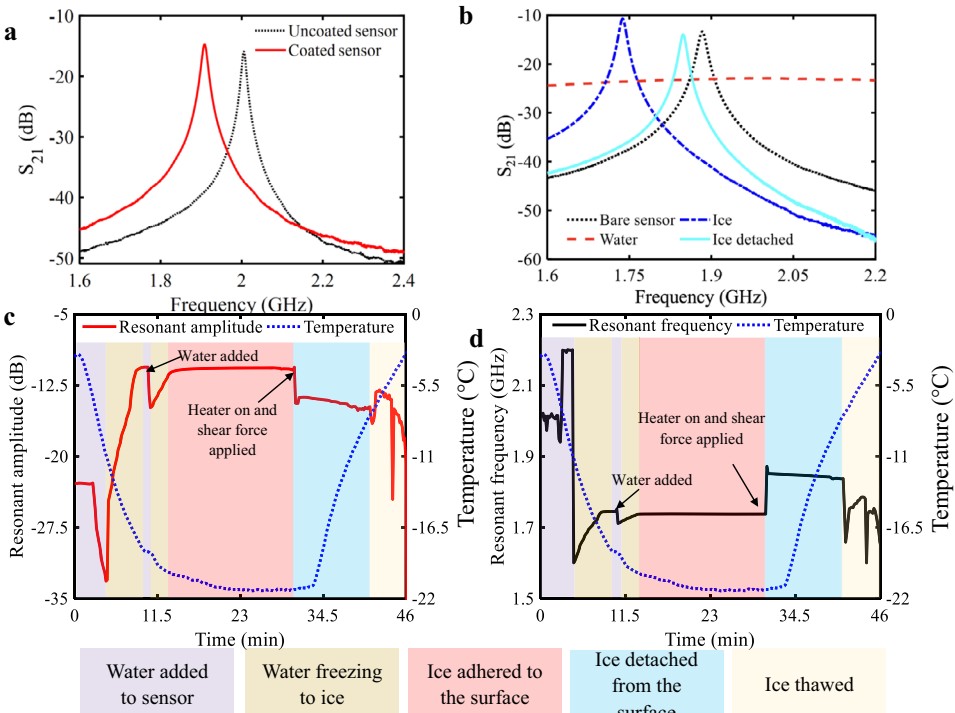

**Fig. 6 | Resonant sensor's response. a** The S$_{21}$ spectrum response of the sensor with and without the LIT material covering the split ring resonator (SRR). S$_{21}$ is the transmitted power from port 1 to port 2 (see Fig. 7). **b** The smart hybrid LIT de-icing system's sensor response to water, adhered ice, detached ice, and the bare sensor. **c** Recorded resonant amplitude and **d** resonant frequency vs. time depicting the water being frozen to the LIT coating covering the sensor, heating the surface locally from $T$ to $T_H$, detaching the ice at $T_H$ with a shear force, and then thawing the system. $T$ is the overall system temperature and $T_H$ is the temperature at the surface of the heater/LIT coating. Source data are provided as a Source Data file[67].

The optimized sensor was used to successfully monitor the entire icing and de-icing process through changes in the sensor's electrical characteristics, such as the resonant frequency (Fig. 6b). Initially, the sensor detected the unfrozen water that was pipetted into the 3D-printed ice mold, through the complete disappearance of the resonant profile (Fig. 6c, d). This detection was possible due to the high permittivity (90) and loss factor (0.3) of water. As the water froze, the resonant profile was recovered due to the much smaller dielectric properties of ice (permittivity of 3.2 and loss factor of 0.001) compared to water. A shift from the baseline (bare sensor) of 0.138 GHz in the resonant frequency and −2.76 dB in the resonant amplitude was observed as the water froze above the sensor. Additional water was then added to achieve the desired thickness of ice, causing a change in the effective dielectric constant of the sensor's environment and resulting in an additional drop of 0.034 GHz in the resonant frequency and −4.26 dB shift in the resonant amplitude. This two-step ice formation can be observed in the sensor's measured response (Fig. 6b-c), and demonstrates that the sensor can detect complex ice compositions including mixtures of water and ice (a common form of precipitate).

After the water was completely frozen on the coated sensor, as verified by the non-changing resonant profile, the heater was engaged (again 4.4 V for 30 s) to obtain a local temperature of $T_H = −5\,°C$. Due to the horizontal configuration of our icing setup (see Methods), even after interfacial fracture the detached ice sits on the surface unadhered, with a small gap in between the ice and LIT coating remaining due to the intrinsic roughness of the UHMW-PE and the imperfect fracture surface. The difference between adhered and unadhered ice was also detectable, as evidenced by the sudden change in the sensor's measured resonant amplitude (−10.61 dB to −14.43 dB) and frequency (1.734 GHz to 1.872 GHz). In more realistic de-icing scenarios environmental forces such as gravity, wind shear, drag, etc. would completely remove the ice from the surface, and even more precise detection

would be possible as the sensor would return to its baseline profile. Finally, the system was thawed, and the sensor's resonant profile began to disappear, indicating the presence of liquid water on the surface (Fig. 6c, d). The spectrums of the bare resonator, water, ice, and detached ice were all distinct and distinguishable. Overall, the smart hybrid LIT system was able to monitor and detect ice formation and de-icing in real time, even with the microwave sensor embedded underneath the LIT coating, *i.e.* contactless detection where direct contact with the precipitate was not necessary.

While the energy efficiency of a de-icing system is highly specific to the application and environmental conditions, here our smart hybrid LIT de-icing system reduces energy consumption in four synergistic ways. First, the use of LIT materials enables mechanical de-icing and therefore eliminates the de-icing step requiring the largest energy input: the phase change of solid ice to liquid water. Second, the mechanism of LIT materials allows for our resistive heaters to be sparsely placed on the surface, requiring only ~10% of total areal coverage to be effective. Third, the meandering traces of copper used to construct the heaters display an order of magnitude improvement in resistance over current heating systems, leading to faster de-icing at smaller applied voltages. And finally, the inclusion of an ice sensor enables the smart system to only be activated when ice is truly adhered to the surface, and also allows the system to be turned off immediately following de-icing.

In this work, we explored a hybrid de-icing system utilizing LIT coatings, where thermal modulation substantially improved the de-icing performance without melting the interface. The interfacial toughness between ice and UHMW-PE was found to be 2.2 times larger at −30 °C than at −5 °C. Accordingly, resistive heaters were patterned periodically underneath the LIT material and were optimized to locally raise the temperature to a warmer but still sub-zero value. Heaters underneath only ~10% of the total ice length were effective at reducing the interfacial toughness, as if the entire surface were held at this

temperature even though 90% of the surface wasn't heated. The hybrid LIT de-icing system was also rendered smart through the addition of a microwave resonator sensor. The sensor operated at a resonant frequency, a resonant amplitude, and a quality factor of 2.005 GHz, −12.95 dB, and 205, respectively, taking advantage of the significant differences in dielectric properties between ice and water at the resonant frequency. This planar, contactless device embedded underneath the LIT material was able to monitor and detect ice formation and removal in real-time. Given the fatal consequences of aircraft icing, and the strong push towards renewables such as wind power, our smart hybrid LIT de-icing system may find immediate usage in multiple ice-prone sectors worldwide, especially considering its energy efficiency, scalability, and durability.

## Methods

### Materials

The LIT film was ultra-high molecular weight polyethylene (UHMW-PE) with an adhesive backing and a thickness of 0.127 mm (McMaster Carr, Catalog No. 1441T11), used as received. The planar microwave sensor and the PCB resistive heaters were fabricated on Rogers RT/Duroid® 5880 Laminates, with permittivity of 2.2, loss tangent of 0.0009, dielectric thickness of 0.79 mm, and copper cladding thickness of 35 μm (Rogers Corporation, Ltd.).

### Characterization of LIT coating's mechanical properties

The storage and loss moduli of the UHMW-PE were measured using a Dynamic Mechanical Analyzer (TA Instruments) at an applied frequency of 1 Hz using a temperature sweep from −60 °C to +30 °C. The strain was not constant during the temperature sweep as the instrument was setup in auto strain adjustment mode. However, the strain variation measured was negligible (0.05% to 0.03%) and within the linear viscoelastic range.

### LIT coating preparation

For experiments where the heater was not involved, the UHMW-PE film was adhered uniformly to an aluminum (Al) sheet with a thickness of 0.254 mm (McMaster Carr, Catalog No. 9708K58) using the adhesive backing. For experiments involving the heater, the UHMW-PE film was adhered directly to the fabricated device (heat and sensor), again using the adhesive backing. The topography of the UHMW-PE surfaces on the Al or the heater/sensor was measured using a LEXT™ OLS5100 3D Laser Scanning Microscope (Supplementary Fig. S9).

### Ice adhesion measurement without heater

The force necessary to de-ice the LIT surface was measured using a custom push-off method described elsewhere[46]. Ice cubes were formed on the surface of the UHMW-PE film using 3D-printed polylactic acid (PLA) molds of various lengths ($L_{ice}$ = 5–200 mm). First, the molds were filled with deionized water at room temperature. Next, the temperature of the surface was decreased to the target temperature using the Peltier stage, and the water was allowed sufficient time to fully freeze (minimum 1 h). Once frozen, a moving probe with a motorized linear stage was connected to a force gauge (NEXTECH, DFS500). The force gauge probe with cross section of 5 mm × 10 mm impacted the ice's mold at a constant speed of 100 μm/s, and the detachment force was measured with 0.1 N accuracy. The ice adhesion measurements were conducted at varying temperatures (−40 °C to −5 °C). The temperature of the ice and surface of the coating was monitored using a BK Precision 725 thermocouple with an accuracy of ±0.7 °C. After each measurement, the coating was cleaned with isopropyl alcohol (VWR International) using a Kim wipe (KimTech).

The ice adhesion strength ($\tau_{ice}$) and interfacial toughness with ice (Γ) are important parameters measured to fully characterize the interface between a surface and ice[46]. In the strength-controlled regime of fracture, the force to dislodge ice ($F_{ice}$) is used to measure

$\tau_{ice}$ using the interfacial area, $A$, or $\tau_{ice} = F_{ice}/A$. In the toughness-controlled regime of fracture that occurs for longer interfaces, this force plateaus at some critical value, $F_c$. One can calculate the toughness of the ice/coating interface using the measured $F_c$ value, the modulus of ice, $E$, and the ice thickness, $H_{ice}$, as $F_c = \sqrt{\Gamma E H_{ice}}$[46]. The interfacial length where the fracture transitions from strength- to toughness-mediated fracture is commonly referred to as the critical length, $L_c$. All of these parameters may directly depend on temperature, or indirectly due to temperature-dependent material properties. Accordingly, the ice adhesion strength and interfacial toughness of UHMW-PE with ice were measured from −5 °C to −40 °C and using ice lengths from 5 to 200 mm (Supplementary Figs. S1, S3).

The calculation of $L_c$, $\tau_{ice}$, and Γ from the $F_{ice}$ versus length measurements was calculated in the following way. An initial guess for the strength and toughness regimes was selected visually, such that the strength data was roughly linear and the toughness data was roughly constant. To determine if lengths of ice close to $L_c$ were within the toughness or strength regime, a student's $t$-test was performed between the $F_{ice}$ value of the length of ice in question, and the current $F_c$ population (all $F_{ice}$ values for lengths of ice greater than the one being considered). If the two populations were statistically similar ($p$-value > 0.05), the datapoint was included in the toughness regime, and the $F_{ice}$ value of the next shortest length of ice was considered. This procedure was repeated until the $F_{ice}$ value from the longest piece of ice in the strength-controlled regime was statistically different ($p$-value < 0.05) from the $F_{ice}$ value of the shortest length of ice in the toughness-controlled regime. The adhesion strength was then determined from the slope of best linear fit in the strength regime. The interfacial toughness was calculated using $\Gamma = F_c^2/(E H_{ice})$[46]. $L_c$ was then determined by the intersection of these two lines. Note that, for some experiments the measurement of $F_{ice}$ for longer lengths of ice served as a substitute for directly measuring Γ, and for such cases we assume $F_{ice} = F_c$.

### Sensor and heater design and fabrication

A planar microstrip sensor consists of copper traces, where the structure resonates according to its geometry and shape, creating a Gaussian frequency response. The frequency where the amplitude of the response is maximized is called the resonant frequency. A microwave resonator sensor was designed and characterized to detect the presence or absence of ice and water on the surface using the resonant frequency and amplitude. The significant difference in the dielectric properties between water and ice has recently enabled sensitive and accurate water, frost, and ice detection via planar microstrip resonators[51]. The microwave split ring resonator (SRR) sensor was designed in Ansys High-Frequency Structure Simulator (HFSS, see Fig. 7). The sensor operated at a resonant frequency of 2 GHz, which was selected due to the difference in dielectric properties of water and ice at this frequency. Further, the sensing structure was optimized to exhibit a sharp bandpass response. The resonant frequency of a microstrip line is governed by the length of the SRR calculated using Eq. (2):

$$l = \frac{c}{2 * f_{res} \sqrt{\varepsilon_r}} \tag{2}$$

here $c$ is the speed of light ($3 \times 10^{11}$ mm/s), $f_{res}$ is the resonant frequency (2 GHz), and $\varepsilon_r = 2.2$ is the relative permittivity of the microstrip line. The calculated length of the SRR at 2 GHz was 50.7 mm. However, since the capacitances between the feedline, the SRR, and the split ring gap influence the resonant frequency, the length of the resonator was optimized in HFSS to 61.6 mm to achieve the desired resonant frequency (see Fig. S7). The dimensions of the final sensor design are shown in Fig. 7.

The PCB heaters are simple traces of copper arranged in a small, confined space with a resistance chosen to output the desired amount

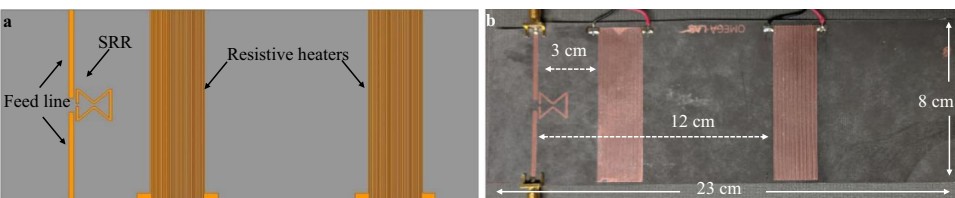

**Fig. 7 | Dimension and spacing of heaters and sensor. a** Split ring resonator (SRR) sensor and resistive heaters modeled in high-frequency structure simulator (HFSS). **b** Optical image of fabricated SRR sensor and resistive heaters.

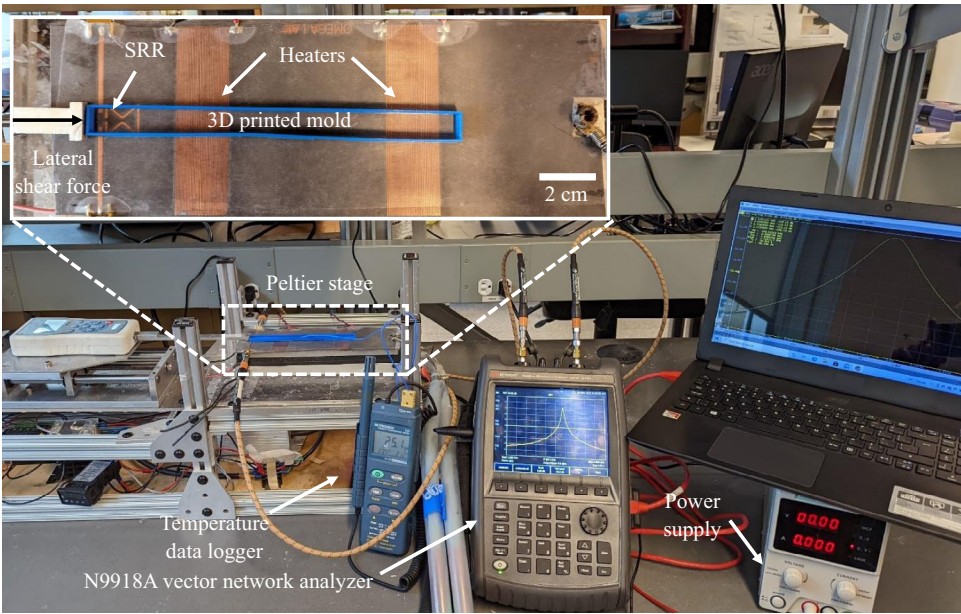

**Fig. 8 | Sensor experimental setup.** The experimental setup consisted of a vector network analyzer, power supply, cold Peltier stage, temperature data logger, printed circuit board heaters, and the split ring resonator sensor.

of heat. A resistive heater with a resistance of 5.9 Ω was additionally patterned on the substrate to provide the active de-icing capabilities. The heaters were placed at a distance of 3 cm and 12 cm from the sensitive region of the sensor (Fig. 7). The sensor and the heater were fabricated following standard PCB fabrication methodologies[66].

### Smart, active de-icing using the LIT coating

The sensor response was monitored with a Keysight Technologies N9918A Vector Network Analyzer (VNA, see Fig. 8). The VNA measures the S-parameters of the microwave sensor over a wide range of frequencies. Similar to the ice adhesion testing on the bare LIT material, deionized water was poured in a 3D-printed mold on the surface of the LIT material and frozen at −25 °C (Fig. 2). The temperature of the ice frozen directly above the heater was also measured using a thermocouple placed within the ice. To do so, the mold was half-filled with water, the thermocouple was inserted, and the water was allowed to freeze. Once the thermocouple was secured in the frozen ice and the temperature stabilized, the heater was engaged to raise the surface temperature locally around the heater from −25 °C to −5 °C. Similar to the testing procedure above, the de-icing performance was measured while engaging the heater and detecting the ice via the sensor. We define the de-icing time ($t_D$) as the amount of time required for the heaters to increase the surface temperature to a target temperature, either −5 °C or −10 °C in this work. Note, however, that without the application of the external mechanical load, simply raising the temperature to −5 °C does not de-ice the surface. Once at the target temperature, the force gauge probe impacted the adhered ice within the mold, and the

detachment force was measured. The temperature of the ice 0.5 mm above the heater was also measured as −25 °C throughout the experiment, confirming that the heating was indeed localized to the surface. While finding the correct operating voltage, the force probe was not engaged, and the ice filled mold was left intact.

### Durability and Scalability Analyses

**Cyclic icing/de-icing.** The LIT coating was applied on the surface of the heater panel and icing/de-icing cycles were performed using different lengths of ice (60–150 mm). The de-icing force was again measured at −20 °C using a force gauge with accuracy of 0.1 N. At least five repeats were performed for each ice length. Following this, 13 additional icing/de-icing cycles were performed on this same sample using 150 mm lengths of ice. Finally, the five repeats of the 60, 80, and 100 mm pieces of ice were cyclically iced/de-iced, for a total of 43 repeated measurements on the same LIT/heater sample.

**Outdoor exposure.** The hybrid de-icing system was placed outdoors for 3 weeks in Toronto, ON, Canada, which included daily thermal fluctuations and one heavy rainstorm event on May 21, 2022. Afterwards, the de-icing force was measured using several lengths of ice at −20 °C.

**Surface contamination.** The LIT coating was contaminated by pipetting acetone onto its surface and allowing it to evaporate (see supporting information, Fig. S5b). The de-icing force for various ice lengths was then measured at −20 °C, with at least five repeats for each length.

**Mechanical Abrasion.** The LIT coating was abraded using 800-grit silicon carbide electrocoated sandpaper (Alibaba Group, China). A power sander (RYOBI 1/3 Corded Sheet Sander, China) was used to constantly abrade the material at 12,000 rpm for 15 min. The roughness and topography of the coating before and after abrasion were measured using a LEXT™ OLS5100 3D Laser Scanning Microscope (Fig. S4). After abrasion, the de-icing force of the coating was measured at −20 °C for various lengths of ice (Fig. S5a).

**Large-scale de-icing using multiple heaters.** To evaluate the scalability of our hybrid de-icing system, a scaled-up version was designed (see supporting information, Fig. S6a). All large-scale de-icing tests were run in a walk-in freezer (Climate Lab, KITE, at the University Health Network, Toronto, Canada) where the average temperature of the room during our 3 days of testing was −18 ± 1 °C with a RH of 75 ± 5%. Using the same heater fabrication method as above, four identical heater panels with dimensions of 24 mm × 80 mm (W × L) were prepared. The panels were then installed on an Al sheet attached to wooden supports (see supporting information, Fig. S6b). The surface of the four panels was coated with one single UHMW-PE film measuring 80 mm by 960 mm. Each heater was connected to a separate power supply set at the optimized voltage. To form a large piece of ice a silicone rubber mold with prescribed internal dimensions of 920 mm × 20 mm × 20 mm (L × W × H) was prepared and placed on the LIT coating. Next, the rubber mold was filled with deionized water and left to fully freeze. Once fully frozen, the rubber mold was removed and a 3D-printed guard was placed around the front end of the ice, such that the force probe tip did not contact the ice directly. The de-icing force was then recorded using the same moving stage and force gauge as above. These tests were repeated at least 5 times. For an ice length of 500 mm the de-icing process was recorded on video, enabling us to monitor the crack propagation front in real time (see supporting movie S1). Space limitations within the freezer prevented recording the larger, 920-mm-long ice detachment process, but the results were visually similar.

## Data availability

Source data are provided with this paper[67].

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

## Acknowledgements

The authors thank the Syilx Okanagan Nation for use of their unceded territory, the land on which the research was conducted. This work was partially supported by the Department of National Defence, under

project CP-3325 allocated to K.G. and M.Z., and the Canada Foundation for Innovation, under grant 41543 allocated to K.G.

## Author contributions

K.G. and M.H.Z. conceived the project. M.C.J. and R.K. are responsible for theoretical design and simulation of sensor and heater. Z.A.D. and M.C.J. performed the experimental work and wrote the manuscript. K.G. and M.H.Z. directed the project. All authors discussed the results and contributed to the manuscript.

## Competing interests

The authors declare no competing interests.
