## [Peer Review File · Nature Communications]

Smart low interfacial toughness coatings for on-demand de-icing without meltingREVIEWER COMMENTS

Reviewer #1 (Remarks to the Author):

In this paper, the authors reported a smart low interfacial toughness coatings for on-demand deicing without melting. By combining the LIT coating with electric heating, substantial breakthroughs have been made for the engineering application. Nevertheless, there are still some problems need to be solved.

1. How about the performance of this technique after cyclic icing-deicing test? For the engineering application, only one deicing test is not enough.
2. How about the long-term outdoor performance? Because the time is limited, how about the performance after outdoor exposure of 7-14 days?
3. For the engineering application, the surfaces are inevitably contaminated. How about the performance if the surface was contaminated?
4. A large scale sample (1m*1m) had better be fabricated. The authors had better demonstrate that large sample has same performance as predicted.

Reviewer #2 (Remarks to the Author):

The authors developed a passive and active de-icing coating by the combination of low interfacial toughness (UHMW-PE), printed circuit board heaters, and ice detecting, microwave sensor. The coating showed temperature dependent interfacial toughness with ice and removed the ice without melting.

I suggest minor revision

- i. The abstract should be short and concise.
- ii. What is the effect of embedded microwave sensor on the coating's interfacial toughness? The interfacial toughness decreases or increases after embedding the sensor.
- iii. What is the composition of the sensor?
- iv. Describe the wettability of the coating before and after embedding the sensor.
- v. What is the durability of the developed coating?
- vi. There should be some supporting videos to show the design and performance of the developed coating.
- vii. The figure 3 should be labelled (a,b,c) with single format. The label c should be outside the graph.
- viii. The conclusion should be short and concise.
- ix. The article about passive anti-icing and active de-icing coating should be cited Journal of Bionic Engineering, 18, 548-558 (2021).

Reviewer #3 (Remarks to the Author):

This is interesting paper showing the hybrid de-icing. Technical results are presented but more scientific approach would be needed. Comparison between different de-icing methods could be described and connected to the importance and impact of the work.

Introduction needs more discussion with the references. Especially, active and passive de-icing and reduction of ice adhesion should be referred with the references. Findings here would need more support from the literature as well as deeper explanations.

Experiments should be described in details. Also, coating used sounds more like a film. Maybe this could be called as film better than coating.

Add more discussion about importance and impact of this work and explain more about comparison between hybrid de-icing and heating. It would be good to connect them to energy consumption, time required or other effecting issues.

"Accordingly, completely melting the ice underneath the heater actually reduced the efficacy of the LIT coating. Moreover, melting is quite energy intensive as the latent heat of melting requires addition energy consumption beyond raising the temperature to 0 °C." This statement should be proofed here in the paper.

Article under revision for: *Nature Communications*
Manuscript Title: Smart low interfacial toughness coatings for on-demand de-icing without melting
Reference number for the article: NCOMMS-22-12650
Authors: Azimi Dijejin, Zahra; Chhajer Jain, Mandeep; Kozak, Ryan; Zarifi, Mohammad H; Golovin, Kevin

We would like to thank the editor and all reviewers for their useful and insightful comments. They have helped us significantly improve our manuscript. The following is a detailed response to all reviewers' comments, reproduced in black. **Our detailed responses are below in red font:**

Reviewer #1 (Remarks to the Author):

In this paper, the authors reported a smart low interfacial toughness coatings for on-demand de-icing without melting. By combining the LIT coating with electric heating, substantial breakthroughs have been made for the engineering application. Nevertheless, there are still some problems need to be solved.

1. How about the performance of this technique after cyclic icing-deicing test? For the engineering application, only one de-icing test is not enough.

Thank you for your comment and for reviewing our manuscript. We've now performed cyclic icing/de-icing tests using several lengths of ice. The following is added to our methods:

“Cyclic icing/deicing

The LIT coating was applied on the surface of the heater panel and icing/de-icing cycles were performed using different lengths of ice (60 – 150 mm). The de-icing force was again measured at -20 °C using a force gauge with accuracy of 0.1 N. At least 5 repeats were performed for each ice length. Following this, 13 additional icing/de-icing cycles were performed on this same sample using a 150 mm length of ice. Finally, the 5 repeats of the 60, 80, and 100 mm length pieces of ice

were cyclically iced/de-iced, for a total of 43 repeated measurements on the same LIT/heater sample.”

The following sentences have been added to the Result and Discussion section:

“In order for our designed hybrid de-icing system to find real-life usage, its performance must be consistent, durable, and scalable. In terms of consistency, the system was exposed to 43 repeated cycles of icing/de-icing. Initially, the ice detachment force for different ice lengths ($L > L_c$) was measured while the heater locally raised the interfacial temperature from $-20\text{ }^{\circ}\text{C}$ to $-5\text{ }^{\circ}\text{C}$ (Fig. 7a). The critical detachment force for the first set was $131 \pm 21\text{ N}$, corresponding to an interfacial toughness with ice of $\Gamma = 1.5 \pm 0.4\text{ J/m}^2$. Additional icing/de-icing cycles were then conducted using 150 mm lengths of ice, followed by a repeat of the initial characterization. After these 43 icing/de-icing cycles, the average de-icing force was statistically equivalent (p -value: 0.22) to its initial value. The surface roughness was also unaffected (Fig. S4 supporting information), indicating that the process of icing and de-icing did not damage the surface.

Fig. 7a Cyclic icing/de-icing tests for ice lengths of 60 – 150 mm ($L > L_c$). F_{ice} values before and after the icing/de-icing cycles are statistically equivalent (p -value: 0.22).”

2. How about the long-term outdoor performance? Because the time is limited, how about the performance after outdoor exposure of 7-14 days?

Our LIT material was placed outdoors for three weeks during the end of Winter in Toronto, ON, Canada. Afterwards, the de-icing force was measured using various lengths of ice (within the toughness-controlled regime of fracture). The following has been accordingly added to our Methods:

“Outdoor exposure

The hybrid de-icing system was placed outdoors for 3 weeks in Toronto, ON, Canada, which included daily thermal fluctuations and one heavy rainstorm event on May 21, 2022. Afterwards, the de-icing force was measured using several lengths of ice at -20 °C.”

And the following sentences were added to our results and discussion section:

“The de-icing force for various ice lengths in the toughness-controlled regime of fracture was also recorded after mechanical abrasion, chemical contamination, and outdoor exposure for three weeks (Fig. 7b, see Fig. S7 for full datasets). The UHMW-PE coating maintained its low ice detachment force for both outdoor exposure and surface contamination (p-value > 0.22), demonstrating the environmental durability of the hybrid de-icing system.

Fig. 7b The de-icing force or interfacial toughness required to remove various lengths of ice after mechanical abrasion, chemical contamination, and outdoor exposure for three weeks.”

Supplementary Fig. S4 Optical and topography images after durability tests. a UHMW-PE sheet before exposure ($S_q = 1.53 \pm 0.05 \mu\text{m}$) and after **b** Chemical contamination by acetone ($S_q = 1.92 \pm 0.08 \mu\text{m}$); **c** abrasion ($S_q = 3.4 \pm 0.3 \mu\text{m}$); **d** Cyclic icing/de-icing ($S_q = 1.4 \pm 0.2 \mu\text{m}$); and **e** outdoor exposure ($S_q = 1.5 \pm 0.3 \mu\text{m}$). (Top images are optical and bottom images are topography heightmaps).

Supplementary Fig. S7 De-icing force versus ice length of LIT material after durability characterization. a After abrasion with 800 grit sandpaper for 15 minutes; **b** After contamination with acetone residue; **c** After exposure to the outdoors for 3 weeks in Toronto, ON, Canada.”

3. For the engineering application, the surfaces are inevitably contaminated. How about the performance if the surface was contaminated?

We agree with the reviewer that surface contamination can present problems for coatings. To study this, we followed the technique of Zarasvand *et al.* [*Adv. Mater. Interfaces* 2101402 (2021)] and allowed acetone to dry on the surface, which naturally contaminates it with organics from the air.

The following figure demonstrates how the surface of a steel sheet gets contaminated using this procedure. Though difficult to visualize, many whiteish deposits form on the surface. Because our

UHMW-PE is also white, steel helps show the contamination better. Note that our outdoor exposure would have also naturally contaminated the surface.

The results of the contamination study are shown in Fig. both Fig. 7b and S7b above. The following was added to our Methods as well:

“Surface contamination

The LIT coating was contaminated following the work of Zarasvand et al.⁵⁴, by pipetting acetone onto its surface and allowing it to evaporate (see supporting information, Fig. S4b). The de-icing force for various ice lengths was then measured at -20 °C, with at least 5 repeats for each length.”

As we explained above in response to your second comment, contamination did not alter the toughness or observed de-icing force statistically significantly (p-value > 0.22).

4. A large scale sample (1m*1m) had better be fabricated. The authors had better demonstrate that large sample has same performance as predicted.

We agree that scalability is the real advantage of LIT materials, and this should be demonstrated. To evaluate the scalability of our hybrid de-icing system, we designed a scaled-up de-icing setup incorporating four independently controlled heaters underneath the LIT material. The details of this experimental setup are now within our Methods:

“Large-scale de-icing using multiple heaters

To evaluate the scalability of our hybrid de-icing system, a scaled-up version was designed (see supporting information, Fig. S5a). All large-scale de-icing tests were run in a walk-in freezer (Climate Lab, KITE, at the University Health Network, Toronto, Canada) where the average

temperature of the room during our 3 days of testing was -18 ± 1 °C with a RH of $75 \pm 5\%$. Using the same heater fabrication method as above, four identical heater panels with dimensions of 24 mm \times 80 mm (W \times L) were prepared. The panels were then installed on an Al sheet attached to wooden supports. The surface of the 4 panels was coated with one single UHMW-PE film measuring 80 mm by 960 mm. Each heater was connected to a separate power supply set at the optimized voltage. To form a large piece of ice a silicone rubber mold with prescribed internal dimensions of 920 mm \times 20 mm \times 20 mm (L \times W \times H) was prepared and placed on the LIT coating. Next, the rubber mold was filled with deionized water and left to fully freeze. Once fully frozen, the rubber mold was removed and a 3D printed guard was placed around the front end of the ice, such that the force probe tip did not contact the ice directly. The de-icing force was then recorded using the same moving stage and force gauge as above. These tests were repeated at least 5 times. For an ice length of 500 mm the de-icing process was recorded on video, enabling us to monitor the crack propagation front in real time (see supporting movie S1). Space limitations within the freezer prevented recording the larger, 920-mm-long ice detachment process, but the results were visually similar.”

Supplementary Fig. S5 De-icing force measurements in walk-in freezer at -20 °C. *a* The ice adhesion setup involving the UHMW-PE coated substrate, heaters, ice, and mechanical detachment instrumentation; *b* Top view of LIT-coated substrate decorated with four heaters with dimensions of 2.54 cm \times 8 cm.”

The following was also added to our results and discussion:

“One advantage of de-icing using LIT materials is their scalability, as the de-icing force is constant for large lengths of ice^{46,47}. To determine if our hybrid de-icing strategy was also scalable, we fabricated a larger-scale system measuring a full meter in length, and utilizing multiple heaters spaced periodically such that only 10% of the surface area was heated (in-line with the results from Fig. 6d,e). The entire setup was then placed inside a walk-in freezer held at -20 °C and iced using a 2-cm-wide piece of ice (Fig. S5). The de-icing force was measured for a 920 mm length of ice when the heaters modulated the local interfacial temperature to $T_H = -5$ °C. Figure 7c shows the de-icing force as a function of ice length for our hybrid de-icing system with heaters underneath only 10% of the total iced area. The de-icing force necessary to dislodge the large-scale ice was statistically equivalent to the values observed for the smaller-scale testing (p-value: 0.08), confirming the scalability of the developed system. Scattering of the light between the LIT material and ice during interfacial separation also allowed us to monitor the crack propagation in real time (Fig. 7d). After ~ 6 seconds the stored strain energy within the interface was released, and the surface was cleanly de-iced with no adhered residue remaining.”

Fig. 7c De-icing force for multi-heater hybrid de-icing system, up to a length of 920 mm. The inset shows the accreted and de-iced surfaces. **d** Movie stills depicting the interfacial crack propagation and adhesive fracture underneath ice with a length of 500 mm and a width of 2 cm. All tests in **a-d** were conducted with 2.54 cm wide heaters locally raising the temperature -20 °C to -5 °C.”

Reviewer #2 (Remarks to the Author):

The authors developed a passive and active de-icing coating by the combination of low interfacial toughness (UHMW-PE), printed circuit board heaters, and ice detecting, microwave sensor. The coating showed temperature dependent interfacial toughness with ice and removed the ice without melting.

I suggest minor revision

i. The abstract should be short and concise.

We appreciate the reviewer's insight on this, and have revised the abstract to make it more concise.

“Ice accretion causes problems in vital industries and has been addressed over the last decades with either passive or active de-icing systems. This work presents a smart, hybrid (passive and active) de-icing system through the unique combination of a low interfacial toughness (LIT) coating, mini printed circuit board heaters, and an ice detecting microwave sensor. The coating's interfacial toughness with ice was found to be temperature dependent and could be modulated using embedded heaters. For example, the coating's interfacial toughness at -30 °C decreased from 5.7 to 0.9 J/m² when the heaters held just 10% of the surface at -10 °C. The synergistic combination of the LIT material and periodic heaters resulted in a de-icing power density 100× greater than a regular, full-coverage heater system. The hybrid, smart de-icing system also showed excellent durability towards repeated icing/de-icing, mechanical abrasion, outdoor exposure, and chemical contamination. Scalability was demonstrated using large (~ 1 m) iced interfaces. Further, a microwave sensor was designed to precisely detect the presence or absence of water or ice on the surface while operating beneath the LIT coating. Using a shift of 0.138 GHz in the resonant frequency and -2.76 dB in the resonant amplitude, the noncontact embedded sensor successfully monitored the status of the LIT coating wirelessly throughout the entire icing/de-icing process. Overall, smart LIT materials offer a paradigm shift in de-icing that can efficiently render a surface ice-free without the need for energetically expensive interface melting.”

ii. What is the effect of embedded microwave sensor on the coating's interfacial toughness? The interfacial toughness decreases or increases after embedding the sensor.

Because the sensor is planar and underneath the LIT material, it essentially has no effect on any properties of the coating, including its interfacial toughness with ice. Figure S1 shows the topography of the sensor and heaters covered by the LIT material. The root-mean-squared roughness of the bare UHMW-PE, and UHMW-PE above the sensor and heaters, was $1.53 \mu\text{m}$, $1.50 \mu\text{m}$, and $1.51 \mu\text{m}$, respectively. Note that the critical length of the UHMW-PE used in this work was found to be $> 8 \text{ cm}$. Accordingly, the interfacial toughness cannot directly be measured above just the sensor, as its physical size would make the fracture of the ice adhered to the UHMW-PE covering the sensor in the strength-controlled regime. However, measurements where the sensor was and was not underneath a portion of the total ice length were statistically equivalent.

Supplementary Fig. S1 Optical and topography images. a UHMW-PE sheet (Root-mean-squared roughness, $S_q = 1.53 \mu\text{m}$); **b** Mini PCB heater covered by the UHMW-PE material ($S_q = 1.51 \mu\text{m}$); **c** The microwave sensor covered by the UHMW-PE coating ($S_q = 1.50 \mu\text{m}$).

iii. What is the composition of the sensor?

The resonant sensor used in this work is made of 35 μm -thick copper that is formed by an etching process on a microwave-opaque substrate. As now stated in the main manuscript:

“The planar microwave sensor and the mini PCB resistive heaters were fabricated on Rogers RT/Duroid® 5880 Laminates, with permittivity of 2.2, loss tangent of 0.0009, dielectric thickness of 0.79 mm, and copper cladding thickness of 35 μm (Rogers Corporation, Ltd).”

These additional details of the sensor and heater composition were added in the Materials and Methods sections for clarity.

iv. Describe the wettability of the coating before and after embedding the sensor.

The wettability of the surface before and after embedding the sensor is shown below. Note though that the sensor area represents a very small percentage of the total surface area, so any small changes in wettability, roughness, etc. caused by the presence of the sensor are highly localized. The water contact angle was measured on both bare UHMW-PE film ($\theta_{Static}: 84 \pm 2^\circ$, $\theta_{Adv}: 96 \pm 0.8^\circ$ and $\theta_{Rec}: 54 \pm 4^\circ$) and sensor coated with the film ($\theta_{water}: 86 \pm 3^\circ$, $\theta_{Adv}: 97 \pm 3^\circ$ and $\theta_{Rec}: 48 \pm 4^\circ$). Statistically the wettability of the surface was not change when the sensor was embedded (p-values of 0.15, 0.71 and 0.08 for static, advancing, and receding water contact angles, respectively).

v. What is the durability of the developed coating?

We've now performed several durability tests to understand the long-term de-icing performance of the designed hybrid system. These include 43 repeated icing/de-icing cycles, mechanical abrasion, chemical contamination, and outdoor exposure. Overall the performance was well-

maintained, so we believe the system is relatively robust. As UHMW-PE is known to be an abrasion resistant, chemically inert material, this was somewhat expected. Added details in our Methods section:

“Durability and Scalability Analyses

Cyclic icing/deicing

The LIT coating was applied on the surface of the heater panel and icing/de-icing cycles were performed using different lengths of ice (60 – 150 mm). The de-icing force was again measured at -20 °C using a force gauge with accuracy of 0.1 N. At least 5 repeats were performed for each ice length. Following this, 13 additional icing/de-icing cycles were performed on this same sample using a 150 mm length of ice. Finally, the 5 repeats of the 60, 80, and 100 mm length pieces of ice were cyclically iced/de-iced, for a total of 43 repeated measurements on the same LIT/heater sample.

Outdoor exposure

The hybrid de-icing system was placed outdoors for 3 weeks in Toronto, ON, Canada, which included daily thermal fluctuations and one heavy rainstorm event on May 21, 2022. Afterwards, the de-icing force was measured using several lengths of ice at -20 °C.

Surface contamination

The LIT coating was contaminated following the work of Zarasvand et al.⁵⁴, by pipetting acetone onto its surface and allowing it to evaporate (see supporting information, Fig. S4b). The de-icing force for various ice lengths was then measured at -20 °C, with at least 5 repeats for each length.

Mechanical Abrasion

The LIT coating was abraded using 800-grit silicon carbide electrocoated sandpaper (Alibaba Group, China). A power sander (RYOBI 1/3 Corded Sheet Sander, China) was then used to constantly abrade the material at 12,000 rpm for 15 min. The roughness and topography of the coating before and after abrasion were measured using a LEXT™ OLS5100 3D Laser Scanning Microscope (see Fig. S4). After abrasion, the de-icing force of the coating was measured at -20 °C for various lengths of ice (Fig. S7).”

And the added details in our results and discussion:

“Durability and scalability

In order for our designed hybrid de-icing system to find real-life usage, its performance must be consistent, durable, and scalable. In terms of consistency, the system was exposed to 43 repeated cycles of icing/de-icing. Initially, the ice detachment force for different ice lengths ($L > L_c$) was measured while the heater locally raised the interfacial temperature from $-20\text{ }^\circ\text{C}$ to $-5\text{ }^\circ\text{C}$ (Fig. 7a). The critical detachment force for the first set was $131 \pm 21\text{ N}$, corresponding to an interfacial toughness with ice of $\Gamma = 1.5 \pm 0.4\text{ J/m}^2$. Additional icing/de-icing cycles were then conducted using 150 mm lengths of ice, followed by a repeat of the initial characterization. After these 43 icing/de-icing cycles, the average de-icing force was statistically equivalent (p -value: 0.22) to its initial value. The surface roughness was also unaffected (Fig. S4 supporting information), indicating that the process of icing and de-icing did not damage the surface.

The de-icing force for various ice lengths in the toughness-controlled regime of fracture was also recorded after mechanical abrasion, chemical contamination, and outdoor exposure for three weeks (Fig. 7b, see Fig. S7 for full datasets). The UHMW-PE coating maintained its low ice detachment force for both outdoor exposure and surface contamination (p -value > 0.22), demonstrating the environmental durability of the hybrid de-icing system. Only the harsh abrasion increased the de-icing force statistically significantly (p -value : 0.002; Fig. 7b). This was due to the increase in roughness of the LIT material, from $S_q = 1.55\text{ }\mu\text{m}$ to $3.39\text{ }\mu\text{m}$, and this was statistically significant (Fig. S4). As interfacial toughness represents a strain energy per unit surface area, the commiserate increase in toughness with roughness was expected. Note, though, that the increase in interfacial toughness observed while using the heaters to modulate the local interfacial temperature of the abraded UHMW-PE ($3.4 \pm 0.9\text{ J/m}^2$) was still substantially less than

that of the unabraded UHMW-PE film without heaters ($6.1 \pm 1.2 \text{ J/m}^2$, see Fig. 5c). Accordingly, the hybrid de-icing system can compensate for any mechanical damage by using the heaters to achieve the required toughness value for a given application.

Fig. 7a Cyclic icing/de-icing tests for ice lengths of 60 – 150 mm ($L > L_c$). F_{ice} values before and after the icing/de-icing cycles are statistically equivalent (p -value: 0.22). **b** The de-icing force or interfacial toughness required to remove various lengths of ice after mechanical abrasion, chemical contamination, and outdoor exposure for three weeks.

Supplementary Fig. S4 Optical and topography images after durability tests. **a** UHMW-PE sheet before exposure ($S_q = 1.53 \pm 0.05 \mu\text{m}$) and after **b** Chemical contamination by acetone ($S_q = 1.92 \pm 0.08 \mu\text{m}$); **c** abrasion ($S_q = 3.4 \pm 0.3 \mu\text{m}$); **d** Cyclic icing/de-icing ($S_q = 1.4 \pm 0.2 \mu\text{m}$); and **e** Outdoor exposure ($S_q = 1.5 \pm 0.3 \mu\text{m}$). (Top images are optical and bottom images are topography heightmaps).

Supplementary Fig. S7 De-icing force versus ice length of LIT material after durability characterization. *a* After abrasion with 800 grit sandpaper for 15 minutes; *b* After contamination with acetone residue; *c* After exposure to the outdoors for 3 weeks in Toronto, ON, Canada.”

vi. There should be some supporting videos to show the design and performance of the developed coating.

A video showing the performance of the hybrid smart LIT coating de-icing system has now been added to the supporting information section. This was for a 50 cm length of ice and was conducted inside a walk-in freezer. Frames from this movie have been included as Fig. 7d, which demonstrates the interfacial crack propagation between the ice and LIT coating:

Fig. 7d Movie stills depicting the interfacial crack propagation and adhesive fracture underneath ice with a length of 500 mm and a width of 2 cm. All tests in **a-d** were conducted with 2.54 cm wide heaters locally raising the temperature $-20\text{ }^{\circ}\text{C}$ to $-5\text{ }^{\circ}\text{C}$.”

vii. The figure 3 should be labelled (a,b,c) with single format. The label c should be outside the graph.

This has been corrected.

viii. The conclusion should be short and concise.

We have revised the conclusion for conciseness:

“In this work, we explored a hybrid de-icing system utilizing LIT coatings, where thermal modulation substantially improved the de-icing performance without melting the interface. The interfacial toughness between ice and UHMW-PE was found to be 2.2 times larger at -30 °C than at -5 °C. Accordingly, resistive heaters were patterned periodically underneath the LIT material and were optimized to locally raise the temperature to a warmer but still sub-zero value. Short heaters underneath only ~10% of the total ice length were effective at reducing the interfacial toughness, as if the entire surface were held at this temperature even though 90% of the surface wasn't heated. The hybrid LIT de-icing system was also rendered smart through the addition of a microwave resonator sensor. The sensor operated at a resonant frequency, a resonant amplitude, and a quality factor of 2.005 GHz, -12.95 dB, and 205, respectively, taking advantage of the significant differences in dielectric properties between ice and water at the resonant frequency. This planar, contactless device embedded underneath the LIT material was able to monitor and detect ice formation and removal in real-time. Considering the larger environmental forces experienced in ice-prone applications such as flight or wind energy, the results presented here may find immediate usage in improving the efficiency of real-world de-icing systems.”

ix. The article about passive anti-icing and active de-icing coating should be cited Journal of Bionic Engineering,18, 548–558 (2021).

We had missed this reference but it is an excellent study, and we've now cited it as Ref. 44 in our main manuscript.

Reviewer #3 (Remarks to the Author):

1. This is interesting paper showing the hybrid de-icing. Technical results are presented but more scientific approach would be needed. Comparison between different de-icing methods could be described and connected to the importance and impact of the work.

2. Introduction needs more discussion with the references. Especially, active and passive de-icing and reduction of ice adhesion should be referred with the references. Findings here would need more support from the literature as well as deeper explanations.

Thank you for these two comments. We've improved both our introduction and discussion sections to better connect our work with prior studies, highlighting the advantages of our proposed system, and improving the explanation of our experimental results.

Page 3:

“For example, Bustillos et al. fabricated a highly thermally/electrically conductive and flexible graphene foam heater that could raise the interface temperature from -20 °C and start to melt a frozen droplet within 33 sec¹⁹. Rahimi et al. used plasma spray to deposit NiCrAlY on a glass/epoxy composite, and showed that both fine and rough morphologies could produce sufficient heat for de-icing purposes²³. Another active de-icing method used by the aviation industry involves flowing hot bleed engine air through the wings of aircraft. Pellissier et al. characterized such hot air pumping for de-icing and their simulation results show that the heat transfer process is highly complex²⁴. However, all previous active de-icing techniques have required the entire interface to be raised above 0 °C, and accordingly these methods consume considerable energy to de-ice large surfaces such as wind turbine blades, aircraft wings, or boat hulls.”

Page 4:

“ Passive de-icing coatings can also reduce the adhesive bond between ice and the coated substrate without necessarily reducing the ice accretion rate. Polydimethylsiloxane (PDMS) and polytetrafluoroethylene (PTFE) are two materials known for their low surface energy that frequently have been used for passive de-icing³⁰. Due their weak bond with ice, such materials have shown exceptionally low ice adhesion strengths in various coating configurations including thin films^{31,32}, self-assembled monolayers³³, and lubricant infused surfaces^{34,35}. Wang et al.

fabricated silicon-oil infused icephobic coatings that demonstrated a low shear ice adhesion strength for cylindrical ice at -10 °C. Similarly, Hou et al. presented fluorinated PDMS films for significant ice adhesion reduction. High surface energy amphiphilic materials can also reduce a surface's ice adhesion strength substantially by creating a nanometer-scale liquid water surface layer, as the bond between liquid water and solid ice is much weaker than a solid-solid bond³⁵. For example, the absorption of water vapor has been demonstrated for poly(ethylene glycol) blended with PDMS, resulting in a thin water layer that enhances passive de-icing³⁶."

Page 5:

"Hybrid de-icing methods can also employ lubricant infusion. Jamil et al. used silicone lubrication on a candle soot coating as a natural light absorber⁴⁴. In their work, conductive iron oxide nano particles served as a heat dissipator, eventually melting the iced interface."

We also point out on Page 5:

"Considering that the latent heat of melting ice (334 J/g) is about 160× greater than ice's specific heat capacity (2.09 J/g °C), a hybrid de-icing system that could avoid melting would provide substantial energy efficiency benefits."

And finally, on Page 27-28:

"While the energy efficiency of a de-icing system is highly specific to the application and environmental conditions, here our smart hybrid LIT de-icing system reduces energy consumption in four synergistic ways. First, the use of LIT materials enables mechanical de-icing and therefore eliminates the de-icing step requiring the largest energy input: the phase change of solid ice to liquid water. Second, the mechanism of LIT materials allows for our resistive heaters to be sparsely placed on the surface, requiring only ~10% of total areal coverage to be effective. Third, the meandering traces of copper used to construct the heaters display an order of magnitude improvement in resistance over current heating systems, leading to faster de-icing at smaller applied voltages. And finally, the inclusion of an ice sensor enables the smart system to only be activated when ice is truly adhered to the surface, and also allows the system to be turned off immediately following de-icing. Given the fatal consequences of aircraft icing, and the strong push towards renewables such as wind power, our smart hybrid LIT de-icing system may find immediate

usage in multiple ice-prone sectors worldwide, especially considering its energy efficiency, scalability, and durability.”

3. Experiments should be described in details. Also, coating used sounds more like a film. Maybe this could be called as film better than coating.

We agree with the reviewer that the UHMW-PE is a film. We now refer to it as such when discussing the material by itself. When it is covering the heaters and/or sensor though it does feel appropriate to call it a coating, as it covers/coats those components.

We have added many more details to our experimental Methods, including our icing, durability, scalability, and sensing studies. These are all highlighted in the marked version of the updated main manuscript, but for brevity have not been reproduced here.

4. Add more discussion about importance and impact of this work and explain more about comparison between hybrid de-icing and heating. It would be good to connect them to energy consumption, time required or other effecting issues.

These and some additional discussions have now been added in both the introduction and discussion sections of our manuscript:

Page 5:

“Considering that the latent heat of melting ice (334 J/g) is about 160× greater than ice’s specific heat capacity (2.09 J/g °C), a hybrid de-icing system that could avoid melting would provide substantial energy efficiency benefits.”

Page 27:

“While the energy efficiency of a de-icing system is highly specific to the application and environmental conditions, here our smart hybrid LIT de-icing system reduces energy consumption in four synergistic ways. First, the use of LIT materials enables mechanical de-icing and therefore eliminates the de-icing step requiring the largest energy input: the phase change of solid ice to liquid water. Second, the mechanism of LIT materials allows for our resistive heaters to be sparsely placed on the surface, requiring only ~10% of total areal coverage to be effective. Third,

the meandering traces of copper used to construct the heaters display an order of magnitude improvement in resistance over current heating systems, leading to faster de-icing at smaller applied voltages. And finally, the inclusion of an ice sensor enables the smart system to only be activated when ice is truly adhered to the surface, and also allows the system to be turned off immediately following de-icing. Given the fatal consequences of aircraft icing, and the strong push towards renewables such as wind power, our smart hybrid LIT de-icing system may find immediate usage in multiple ice-prone sectors worldwide, especially considering its energy efficiency, scalability, and durability.”

5. “Accordingly, completely melting the ice underneath the heater actually reduced the efficacy of the LIT coating. Moreover, melting is quite energy intensive as the latent heat of melting requires addition energy consumption beyond raising the temperature to 0 °C.” This statement should be proofed here in the paper.

Yes, this statement was poorly worded, and has been heavily edited/rewritten:

“Accordingly, completely melting the ice adhered directly above the heater was actually detrimental to the performance of the LIT coating by causing cohesive fracture.”

REVIEWERS' COMMENTS

Reviewer #1 (Remarks to the Author):

This manuscript has been carefully modified.

Reviewer #2 (Remarks to the Author):

This manuscript is highly innovative and scientific. It is recommended to accept.